# Key Chemokine Pathways in Atherosclerosis and Their Therapeutic Potential

**DOI:** 10.3390/jcm10173825

**Published:** 2021-08-26

**Authors:** Andrea Bonnin Márquez, Emiel P. C. van der Vorst, Sanne L. Maas

**Affiliations:** 1Institute for Molecular Cardiovascular Research (IMCAR), RWTH Aachen University, 52074 Aachen, Germany; anbonninmarq@ukaachen.de; 2Interdisciplinary Center for Clinical Research (IZKF), RWTH Aachen University, 52074 Aachen, Germany; 3Department of Pathology, Cardiovascular Research Institute Maastricht (CARIM), Maastricht University Medical Centre, 6229 ER Maastricht, The Netherlands; 4Institute for Cardiovascular Prevention (IPEK), Ludwig-Maximilians-University Munich, 80336 Munich, Germany; 5German Centre for Cardiovascular Research (DZHK), Partner Site Munich Heart Alliance, 80336 Munich, Germany

**Keywords:** chemokines, chemokine receptors, cardiovascular disease, atherosclerosis

## Abstract

The search to improve therapies to prevent or treat cardiovascular diseases (CVDs) rages on, as CVDs remain a leading cause of death worldwide. Here, the main cause of CVDs, atherosclerosis, and its prevention, take center stage. Chemokines and their receptors have long been known to play an important role in the pathophysiological development of atherosclerosis. Their role extends from the initiation to the progression, and even the potential regression of atherosclerotic lesions. These important regulators in atherosclerosis are therefore an obvious target in the development of therapeutic strategies. A plethora of preclinical studies have assessed various possibilities for targeting chemokine signaling via various approaches, including competitive ligands and microRNAs, which have shown promising results in ameliorating atherosclerosis. Developments in the field also include detailed imaging with tracers that target specific chemokine receptors. Lastly, clinical trials revealed the potential of various therapies but still require further investigation before commencing clinical use. Although there is still a lot to be learned and investigated, it is clear that chemokines and their receptors present attractive yet extremely complex therapeutic targets. Therefore, this review will serve to provide a general overview of the connection between various chemokines and their receptors with atherosclerosis. The different developments, including mouse models and clinical trials that tackle this complex interplay will also be explored.

## 1. Introduction

Cardiovascular diseases (CVDs) are the leading cause of death worldwide with an estimated 17.8 million annual deaths according to the Global Burden of Disease Study 2017 [1]. Atherosclerosis, a pathology in which inflammation is one of the major driving forces of disease progression, has been recognized as the leading cause of CVDs [2,3,4]. Although therapeutic strategies for atherosclerosis have traditionally focused on lipid management or reduction, the recent Canakinumab Anti-inflammatory Thrombosis Outcome Study (CANTOS) revealed promising results with the use of an anti-inflammatory therapy to reduce the incidence of cardiovascular (CV) events without affecting cholesterol levels, demonstrating the potential of anti-inflammatory approaches [5]. Canakinumab targets interleukin-1β (IL-β), which is one of the inflammatory components contributing to the development of atherosclerosis; however, multiple inflammatory components support atherosclerotic development and these components represent various other therapeutic targets that remain to be investigated [6]. The selectivity of inflammatory targets specific to atherosclerosis has more recently been highlighted by the Cardiovascular Inflammation Reduction Trial (CIRT). Treatment with low-doses of the anti-inflammatory drug methotrexate failed to reduce the levels of IL-1β, IL-6 or C-reactive protein (CRP), and CV events in patients with stable atherosclerosis and high CV risk [7]. However, methotrexate was successful in reducing CVD risk in patients with rheumatoid or psoriatic arthritis [8,9,10], which further supports the selectivity of inflammatory targets and emphasizes the importance of selecting the appropriate inflammatory pathways. Besides these cytokines, chemokines and their receptors are also potential targets due to their specific roles in atherosclerotic disease progression [4,11,12,13]. Common consensus points to chemokines being crucial factors in the development of atherosclerosis [4,12,13]. Following trauma or infection, chemokines are important modulators of leukocyte adhesion and recruitment to the arterial endothelium, which is the initial step in the development of atherosclerosis [4,11]. In this review we present a brief overview of atherosclerosis and chemokines, highlighting the main chemokines involved in the pathogenesis, and discuss ongoing developments to translate these targets into therapeutic strategies.

### 1.1. Chemokines

Chemokines, also known as chemotactic cytokines, direct the migration of cells, most notably leukocytes, to sites of infection or trauma. Furthermore, they also play a part in the maintenance and development of certain immune responses and are involved in wound healing, angiogenesis, and various other cellular functions [14,15,16]. Interestingly, chemokines have both homeostatic and inflammatory functions [17,18]. These small signaling proteins (8–12 kDa) operate through interactions with either G protein-coupled receptors (GPCRs) or the less common atypical chemokine receptors (ACKRs) [4,13,19]. Chemokines can be divided into four different subgroups: CC-, CXC-, CX3C-, and XC-, based on the position of the cysteine residues nearest to the N-terminus. For example, CC- chemokines have two consecutive cysteines near the N-terminal while the first two cysteines in CXC-chemokines are separated by one other amino acid [20]. The chemokine systematic nomenclature consists of the subfamily name followed by an “L” (for ligand) and ends with a number representative of its order of discovery. Similarly, chemokine receptors are named according to the subgroup name, an “R” (for receptor), and the number based on its chronological discovery [20,21,22]. The ligands and their receptors are known to lack a certain selectivity, as various chemokines will bind to more than one receptor and a receptor may bind multiple chemokines [20,21,23]. States of prolonged or chronic inflammation are known to result in the continuous recruitment and accumulation of cells mediated by chemokines, resulting in the perpetuation of inflammation, which can lead to various pathologies like atherosclerosis, thereby increasing the risk of CV events [24].

### 1.2. Atherosclerosis

Atherosclerosis is a chronic inflammatory disease, characterized by the formation of atherosclerotic plaques in large and medium-sized arteries [25]. The development of atherosclerosis begins with an injury to the arterial endothelium, followed by the recruitment of inflammatory cells to the injured area [26]. This inflammatory response is in partly regulated via nuclear factor kappa B (NF-κB) following activation by toll-like receptor 4 (TLR4) [27,28]. As a consequence of the trauma and subsequent inflammatory response, endothelial cells (ECs) are activated and endothelial permeability increases, facilitating the infiltration and accumulation of atherogenic lipoproteins, such as low-density lipoprotein (LDL). These LDL particles can be oxidized by reactive oxygen species (ROS) [29], and upon accumulation of oxidized LDL (oxLDL), together with the endothelial damage, an inflammatory response is triggered. This inflammatory response is promoted through an increased expression of adhesion molecules, chemokines, and inflammatory cytokines resulting in further recruitment of inflammatory cells, such as monocytes, to the intima [30,31,32]. The TLR4/NF-κB pathway supports this process as NF-κB activation triggers the expression of adhesion molecules, cytokines and matrix metalloproteinase (MMPs) [27]. Similarly, p38 mitogen-activated protein kinase (MAPK) can be induced by oxLDL to upregulate vascular cell adhesion molecule-1 (VCAM-1) and E-selectins, and CCL2 [33]. Monocytes can then adhere to the endothelium by interacting with adhesion molecules, such as VCAM-1, intercellular adhesion molecule-1 (ICAM-1) and P/E-selectins [30,32]. These are expressed on the activated endothelium, allowing infiltration of the monocytes into the vessel wall, where they eventually differentiate into macrophages [30,32]. The degradation of the extracellular matrix additionally facilitates the migration of macrophages/monocytes into the tissues. MMP-9 is crucial for this degradation and its transcription is regulated by NF-κB [34]. Furthermore, leukocytes can also be recruited by activated platelets, via the release of chemokines, further propagating the inflammatory response [35]. Macrophages present in the vessel wall phagocytize oxLDL and lysosomal acid lipase subsequently hydrolyzes cholesteryl esters in the LDL particle and releases free cholesterol in the macrophages [36]. This free cholesterol becomes esterified in the endoplasmic reticulum of macrophages by acyl coenzyme A: cholesterol acyltransferase-1 (ACAT1). Cholesteryl esters are stored as lipid droplets which are eventually hydrolyzed via neutral cholesterol ester hydrolase (NCEH) and transferred and excreted via cholesterol transporters to high-density lipoprotein (HDL) [37]. In a proinflammatory environment, such as in atherosclerotic lesions, this process is disturbed due to the expression of cholesterol transporters and NCEH being downregulated. Furthermore, the amount of oxLDL as well as the expression of ACAT1 is upregulated, leading to the excessive accumulation of lipid droplets in the endoplasmic reticulum. This in turn results in the formation of foam cells [38,39]. The accumulation of these lipid-rich cells in the intima generates a fatty streak, which following further cell recruitment develops into an atheromatous plaque [40]. Plaque build-up is further exacerbated by foam cells eliciting an inflammatory response via the release of cytokines and chemokines. Hence, leukocytes are recruited and mobilized to the area, supporting the continuation of this circuitous process [19]. Over time, these lipid-filled foam cells die and become part of a necrotic core, triggering the proliferation of vascular smooth muscle cells (VSMCs). VSMCs migrate from the media to the area between the necrotic core and the lumen, where they secrete fibrous elements that stabilize the lesion and create a fibrous cap [30,41]. Plaque formation continues due to the proinflammatory environment, further narrowing the vessel. Eventually, various inflammatory factors and enzymes like collagenases are secreted by foam cells, leading to the weakening of the fibrous cap, making it susceptible to rupture [30]. Finally, rupture of the plaque results in the activation of platelets and the aggregation of which precipitates the occurrence of thrombotic events [35]. The formed clot can occlude the narrowed lumen, leading to ischemia and clinical manifestations of the disease, presenting as myocardial infarction or stroke [19].

## 2. Chemokines, Chemokine Receptors, and Atherosclerosis: A Complex Interplay

Chemokines and their receptors play an important, yet complex role in chronic inflammatory responses related to atherosclerosis. In this section, we will highlight the main chemokines and chemokine receptors that are involved in the initial development, progression, and regression of atherosclerotic lesions.

### 2.1. Initiation of Atherosclerosis

As described above, dysfunction of the arterial endothelium marks the first step in the development of atherosclerosis. This dysfunction leads to the main process in the initiation phase of atherosclerosis, the mobilization and recruitment of leukocytes to the activated endothelium. The notable chemokines and chemokine receptors involved in the initiation of atherosclerosis are highlighted in this section.

Fractalkine, also known as CX_3_CL1, can exert cytotoxic effects on the endothelium, which can result in vascular injury, [42,43,44] a first step in the cascade of events leading to atherosclerotic plaque development (Figure 1, pathway 1). CX_3_CL1 can be present as a membrane-bound molecule as well as in soluble form [45]. This special characteristic allows CX_3_CL1 to behave as an adhesion molecule. Leukocytic CX_3_CR1 can bind to membrane-bound CX_3_CL1 expressed on endothelial cells, and this specific mechanism, together with the binding of natural killer (NK) cells and cytotoxic T lymphocytes, may be responsible for the apparent cytotoxic effect of CX_3_CL1. The CX_3_CL1−CX_3_CR1 interaction activates the lymphocytes, resulting in the release of lytic granules, which damages the endothelium [42,46] (Figure 1, pathway 1). In contrast to this cytotoxic effect of CX_3_CL1, Döring et al. showed that endothelial CXCR4 deficiency results in a disturbed barrier function of the endothelium and thus in increased vascular permeability [47] (Figure 1, pathway 3), which in turn accelerates the development of atherosclerosis. This demonstrates that the chemokine receptor CXCR4 is actually implicated in the protection of the endothelium. Another important player in the activation of the endothelium is oxLDL. As previously mentioned, LDL, which accumulates in the intima, is highly susceptible to oxidation, resulting in the formation of oxLDL. OxLDL contains lysophosphatidylcholine, which is hydrolyzed into lysophosphatidic acid (LPA) by autotaxin [19,48]. Next, LPA can activate ECs, causing them to secrete CXCL1 which acts via the receptor CXCR2 [8,10] (Figure 1, pathway 2). CXCL1 is implicated in the recruitment of both neutrophils and classical monocytes to the site of endothelial dysfunction [11,12,49] (Figure 1, pathway 1). Classical Ly-6C^hi^ monocytes exhibit more proinflammatory phenotype as compared to nonclassical monocytes. This monocyte subset preferentially adheres to the activated endothelium and is therefore predominantly found in atherosclerotic plaques [50,51]. Moreover, monocyte chemotaxis to the arterial wall is stimulated by macrophages secreting macrophage inhibitory factor (MIF), a cytokine with chemokine-like functions that binds to CXCR2 [11] (Figure 1, pathway 1). Additionally, MIF can also bind to CXCR4, and binding to either CXCR2 and CXCR4 results in the promotion of monocyte and T-cell recruitment and the arrest of the rolling leukocytes [52] (Figure 1, pathway 2). However, CXCR2-MIF binding is not monocyte specific, since this interaction is also implicated in neutrophil chemotaxis [52]. Furthermore, Soehnlein et al. established that CCR1 and CCR5 are key mediators in the recruitment of Ly-6C^hi^ monocytes into atherosclerotic lesions [51]. The ligand, CCL5, commonly known as RANTES, is deposited by platelets on the activated endothelium and binds to both CCR1 and CCR5 (Figure 1, pathway 5). The deposition of CCL5 results in leukocyte arrest and allows leukocytes to transmigrate into the arterial intima [53] (Figure 1, pathway 5). Veillard et al. observed that CCL5 is highly expressed in atherosclerotic plaques. They treated mice with the CCL5 antagonist Met-RANTES and observed a considerable decrease in the infiltration of leukocytes accompanied by decreased lesion sizes in aortic roots and thoraco-abdominal aortas as compared to controls [54]. Supporting the atherosclerotic role of CCL5, a deficiency of CCR5 (*Ccr5*^−/−^) in apolipoprotein E deficient (*Apoe**^−/−^**)* mice also resulted in reduced lesion formation [55]. Combined, these studies clearly demonstrate the important role of the CCL5-CCR1-CCR5 axis in leukocyte/monocyte recruitment and early plaque formation (Figure 1, pathway 5). Similar to classical monocytes, CCL5 has been shown to activate neutrophils and stimulate adhesion and extravasation of the endothelium following interaction with its receptors CCR1, CCR2, and CCR5 [56].

Another chemokine that is highly involved in the mobilization of monocytes is CCL2 (Figure 1, pathway 4), also known as monocyte chemoattractant protein-1 (MCP-1), which contributes to plaque development [57]. In animal models, a deficiency of CCL2 or CCR2 resulted in decreased lesion development by reducing the accumulation of lipids and the infiltration of monocytes [58,59,60]. Additionally, studies appear to point towards the CCR2−CCL2 axis being required for the emigration of classical monocytes from the bone marrow into the peripheral circulation [61,62], which could further explain why a deficiency in either the receptor or the chemokine results in less plaque development. More recently, Winter et al. demonstrated the presence of a circadian pattern in the release of CCL2 by myeloid cells in mice [63]. This study investigated and described the rhythmic circadian component of arterial leukocyte recruitment in the early stages of atherosclerosis. Moreover, the CCR2−CCL2 axis was established as a driver behind the rhythmic myeloid cell adhesion as it followed the same oscillatory pattern of CCL2 release and CCR2 expression on classical monocytes and neutrophils [63] (Figure 1, pathway 4). Although CCL5 and CXCL1 and their receptors exert similar effects on myeloid cells, they did not share the same oscillatory expression levels [63].

Besides the above described pathways, chemokines and chemokine receptors also play an important role in atherosclerosis development on other fronts. Under chronic inflammatory conditions, resident stromal mesenchymal cells (MSCs) are activated by proinflammatory molecules, including IL-17, IL-23, LTα_1_β_2_, TNFα, and chemokines, including CCL2, CCL3, CXCL9, CXCL10, and CXCL11 [64,65,66,67]. These molecules are produced by lymphotoxinα1β2-expressing hematopoietic lymphoid tissue inducer (LTi) cells [68]. Activated MSCs can function as lymphoid tissue organizer (TLo)-like cells [69,70]. Medial VSMCs are primed to release lymphorganogenic chemokines, such as CXCL13 and CCL21. These chemokines mediate the recruitment and regulate the infiltration of lymphocytes into the adventitia, leading to the formation of arterial tertiary lymphoid tissues (ATLOs) in the adventitia adjacent to atherosclerotic plaque [68,69,70]. The formation of ATLOs has been observed in humans as well as in aged mice [71,72,73]. These atherosclerosis-associated lymphoid aggregates vary in degree of complexity, ranging from small B/T-cell clusters to well-structured lymph node-like, though unencapsulated, lymphoid tissues [67]. Interestingly, their structure and size correlate with disease severity in a lymphotoxin β receptor (LTβR)-dependent manner [71,73,74]. The effects of B cells, either originating from ATLOs or other lymphoid organs, on the development of atherosclerosis is dependent on the B cell subset, as B-1 cells attenuate atherosclerosis while B-2 cells aggravate the chronic inflammatory process [75,76,77,78,79]. B-1 cells exert their antiatherogenic properties via the secretion of IgM, which inhibits the formation of foam cells, while in comparison, B-2 cells stimulate type 1 T helper (Th1) and dendritic cells to play a proatherogenic role [80]. During atherosclerosis, B cell migration is directed by various sets of chemokines and chemokine receptors. The CXCL12/CXCR4, CXCL13/CXCR5, and CCL19/CCL21/CCR7 pairs support B cell homing to the site of lymphoid structures [81], as CXCL13 and CCL21 are crucial for the recruitment of B cells to ATLOs [71]. The CCL20/CCR6 chemokine pair plays a role in the recruitment of B-1 cells towards the atherosclerosis-prone area of the aorta. This process is controlled by the inhibition of differentiation-3 (Id3), which is known to attenuate diet-induced atherosclerosis [82]. Increased numbers of B-1 cells and increased IgM secreting cells were found in the perivascular adipose tissue (PVAT) of *Apoe*^−/−^ mice with a B cell-specific knockout of the gene encoding for Id3. Furthermore, fat-associated lymphoid clusters in PVAT harbored high numbers of B cells [75].

PVAT may play a crucial role in the development and progression of atherosclerosis considering the proximity between PVAT and the vascular wall. PVAT may particularly intervene with endothelial homeostasis [83] and inflammatory cell recruitment [84]. However, the pathophysiological characteristics of PVAT appear to vary upon the metabolic status [85,86] and anatomical location [87]. In homeostatic conditions, PVAT releases protective endothelium-derived factors with antiatherogenic properties, such as H_2_S, NO, and adiponectin [88]. For example, PVAT-derived adiponectin suppressed perivascular collar-induced carotid atherosclerotic lesion formation in high-fat diet fed *Apoe*^−/−^ mice by increasing macrophage autophagy [89]. On the contrary, a lack of H_2_S and NO initiates atherosclerosis and accelerates the progression of the plaque development [90,91]. Murine PVAT transplantation experiments revealed an increase in PVAT-derived CCL2 production, subsequently promoting atherosclerosis [92], while PVAT-derived low-density lipoprotein receptor-related protein-1 (LPR-1) attenuated lesion development [93]. In humans, dysfunctional PVAT produced inflammatory adipokine and cytokines, such as leptin, TNF-α, and IL-6, which induced the production of endothelial CCL2, ICAM-1, and VCAM-1, enhancing endothelial dysfunction, resulting in monocyte recruitment and promoting vascular lesion formation and progression.

To summarize, following endothelial dysfunction, a chain of events is triggered that recruits various myeloid cells to the activated endothelium via chemokine−chemokine receptor interactions. This cyclic process is self-sustaining as chemokines and other signaling molecules help feed and prolong this inflammatory environment characterized by leukocyte recruitment and leading to the initiation of atherosclerosis.

### 2.2. Atherosclerotic Progression

The roles of chemokines and their receptors in leukocyte mobilization and recruitment allow for the initiation of atherogenesis; however, these small molecules also play a significant role in the progression of atherosclerosis. In this section, we will provide an overview of some of the chemokine and chemokine receptors that play a key role in the advancement of atherosclerosis.

During the course of lesion development, more and more macrophages transform into foam cells or undergo apoptosis [94,95]. These foam cells are unable to phagocytize debris and apoptotic cells. This defective or reduced efferocytosis contributes to the formation of the necrotic core and the continued release of proatherogenic signals by apoptotic cells, which further destabilizes the plaque [94,95]. Interestingly, CX_3_CL1 is directly implicated in efferocytosis, as it is released by apoptotic cells and recruits macrophages to clear the apoptotic debris [96] (Figure 2, pathway 1). This chemokine, in addition to being involved in the adhesion and chemotaxis of leukocytes, has a role in reducing the release of inflammatory cytokines and promotes the expression of both prosurvival and antioxidant genes [96]. The CX_3_CL1−CX_3_CR1 axis is antiapoptotic and provides survival signals to monocytes and macrophages, [45,51,97,98] allowing for the accumulation of these cells and consequently the accumulation of foam cells. Apoptosis and efferocytosis, as they pertain to monocytes/macrophages in atherosclerosis, are extremely complex. Studies point to a beneficial role in the apoptosis and clearance of monocytes and foam cells in earlier stages of atherosclerosis, whereas during the later stages, apoptosis of these monocytes and foam cells can result in disease progression or thrombosis [97,99]. Although most studies appear to show a decrease in plaque development due to deficiencies in the CX_3_CL1−CX_3_CR1 axis [13,42,97,100,101], the described differences based on disease stage have to be taken into account, as inhibition of the survival signals conferred by this chemokine axis at more advanced stages of atherosclerosis may actually exacerbate disease [97]. Similar to CX_3_CL1, CXCL4, also known as platelet factor 4 (PF4), also exerts antiapoptotic effects, specifically on monocytes. This chemokine was not only shown to prolong the survival of monocytes, but also induced their differentiation into macrophages [102]. Accordingly, Sachais et al. showed a reduction of atherosclerotic plaque burden in both C57Bl/6 *Pf4*^−/−^ and *Apoe*^−/−^*Pf4*^−/−^ mice when compared to their respective controls [103]. These results are in line with those from CX_3_CL1 and clearly show the important role these chemokines play in atherogenesis. However, a more detailed understanding of the cell- and time-specific effects of these chemokine interactions is still required.

Aside from CX_3_CL1, CXCL16 is the only other chemokine that exists in both a soluble and transmembrane form [11] and it is expressed by dendritic cells, macrophages, B cells, T cells, SMCs, and ECs [104]. Membrane-bound CXCL16 can act as an adhesion molecule for cells which express CXCR6 [104], contributing to leukocyte accumulation (Figure 2, pathway 2). Additionally, CXCL16 operates as a scavenger receptor for oxLDL and mediates the uptake of such atherogenic lipoproteins, by both macrophages and SMCs, facilitating the formation of foam cells [13,104] (Figure 2, pathway 2). When expressed by ECs, as a transmembrane protein, CXCL16 attracts and binds leukocytes but can also activate platelets following their binding to CXCL16 via CXCR6 [105], thereby promoting platelet aggregation on the endothelium (Figure 2, pathway 2). Continuing the parallels between CXCL16 and CX_3_CL1, the CX_3_CL1−CX_3_CR1 axis has also been found to participate in platelet activation and their adhesion to the endothelium [4] (Figure 2, pathway 2). Platelets are highly involved in hemostasis and thrombosis, a common consequence of ruptured atherosclerotic lesions. In addition, platelets are also implicated in the regulation of various immune responses and the development of atherosclerosis, which is for the most part modulated through chemokines [106,107]. In a study by Postea et al., platelets in both humans and mice overexpressed CX_3_CR1 either in the presence of hyperlipidemia or following platelet activation [108]. The increased expression of CX_3_CR1 allowed for an increased binding of CX_3_CL1, promoting the aggregation of platelets and, in turn, increasing the recruitment of monocytes [108], perpetuating the ideal conditions for atherosclerotic progression. Furthermore, various other chemokines can act as platelet activators, including CXCL12 [12], CCL12 and CCL22 [109,110]. CCL12 and CCL22 act via their receptor CCR4 and have been demonstrated to stimulate the activation of platelets, which results in platelet adhesion and aggregation followed by the release of proinflammatory chemokines perpetuating the cycle of atherosclerosis [109,110]. Meanwhile, CXCL12, also known as stromal cell-derived factor 1 (SDF-1), binds to CXCR4 and exerts similar effects on platelets [12], although CXCL12 also plays a role in various other atherosclerotic processes [13]. CXCL12 is more highly expressed in SMCs, ECs, and macrophages in human atheroma than in nonatherosclerotic vessels [111]. This ligand not only activates platelets, but is also released by activated platelets, and it can bind to both CXCR4 and ACKR3, previously known as CXCR7 [112]. The various effects exerted by CXCL12 are not only cell-specific, but also dependent on which receptor the ligand binds to [112]. For example, platelet derived CXCL12 has been shown to modulate monocyte migration via CXCR4, while binding to ACKR3 appears to be more involved in monocyte adhesion and survival [112] (Figure 2, pathway 3). Additionally, the binding of CXCL12 to either receptor promotes differentiation into macrophages and phagocytosis of platelets that became apoptotic due to the uptake of oxLDL, which in turn induces foam cell formation [112] (Figure 2, pathway 3). Ma et al. investigated the ACKR3 expression in atheroma and identified an increased expression in atherosclerotic *Apoe*^−/−^ mice compared to control. They pointed to the involvement of ACKR3 in atherogenesis via the promotion of macrophage phagocytosis [113] (Figure 2, pathway 3). This atypical receptor can additionally bind CXCL11 and MIF, [106,114] and studies have recently revealed its conflicting roles in atherosclerosis. Interestingly, emerging as a potential cardioprotective receptor, ACKR3 has been shown to be involved in angiogenesis, endothelial proliferation, and the promotion of cholesterol uptake by adipose tissue [106,115]. The role of adipocyte-expressed ACKR3 in the regulation of lipids, was also studied in *Apoe*^−/−^ mice under hyperlipidemic conditions [116]. Increased cholesterol uptake by adipose tissue resulted in a reduction of blood cholesterol levels and led to hyperlipidemia [106,117]. Therefore, ACKR3 seems to play a role in the amelioration of atherosclerosis, since excessive cholesterol levels and hyperlipidemia are known drivers of atherosclerosis.

As previously mentioned, the formation of the necrotic core triggers VSMCs to proliferate and secrete collagen and elastin, thereby stabilizing the plaque via the formation of a fibrous cap [12]. However, certain VSMC phenotypes are associated with atherogenesis. VSMCs deficient in CXCR4 exhibit a synthetic phenotype unable to maintain normal vascular reactivity and contractile responses which is connected to the development of atherosclerosis [47], demonstrating a potential atheroprotective effect of SMC-expressed CXCR4 (Figure 2, pathway 4). Strikingly, MIF also has an important role in VSMCs and is thus involved in the advancement of atherosclerosis [118]. MIF influences the destabilization of the plaque by inhibiting VSMC proliferation and regulating the proteolytic activity and breakdown of elastin and collagen [118], creating more unstable plaques (Figure 2, pathway 4). This was further confirmed by observing a reduced proteolytic capacity of SMCs isolated from the aortas of *Ldlr*^−/−^*Mif*^−/−^ mice [118]. Correspondingly, CXCL10 also plays a role in the destabilization of plaques through its ability to modulate VSMC proliferation and motility [119]. CXCL10, otherwise known as interferon gamma (IFNγ)-inducible protein-10 (IP10), acts via its receptor CXCR3. Furthermore, it is involved in the chemotaxis of leukocytes and has a role on the proliferation as well as the migration of ECs and VSMCs [119]. CXCL10 produced by VSMCs has been demonstrated to act as an inhibitor of endothelial healing via receptor interactions and directly interferes with EC proliferation by competitively binding to endothelial growth factors [120]. This release of CXCL10 by VSMCs occurs as a response to type 1 T helper (Th1) cell cytokine release [120]. Furthermore, *Apoe*^−/−^*Cxcr3*^−/−^ and *Apoe*^−/−^*Cxcl10*^−/−^ mice had significantly reduced atherosclerotic lesion sizes when compared to *Apoe*^−/−^ controls, which coincided with a decreased accumulation of Th1 cells, also known as CD4+ T-cells, in the plaques [119]. Th1 cells are the most abundant Th cell subset in atherosclerotic lesions and the main Th cell subset promoting atherosclerosis [121]. Interestingly, the effect of Th1 and Th2 cells in atherosclerosis can also be highlighted with other chemokine receptors. For example, CCR1 and CCR5 inhibition or blockade exerts contrasting effects on atherogenesis [55,122]. CCR1 deletion favors a Th1 response which results in the change to a proatherogenic phenotype, while CCR5 deletion, favoring a Th2 immune response, displays an atheroprotective phenotype [55,122]. These effects were apparent in an *Apoe*^−/−^*Ccr5*^−/−^ mouse model, which showed upregulation of anti-inflammatory IL-10, a more stable plaque, and a reduction in plaque size and mononuclear cell infiltration when compared to *Apoe*^−/−^*Ccr1*^−/−^ [55,122]. Another important receptor implicated in T-cell recruitment into the aortic wall is CXCR6 [123]. Mice deficient in CXCR6 had a reduced percentage of CXCR6+ T-cells in the aorta, which significantly decreased IFNγ production, reduced macrophage recruitment, and exhibited diminished lesion formation. This reduced secretion of IFNγ additionally regulated the expression of CCL2, CXCL10, and CX_3_CL1, which may also explain reduced leukocyte recruitment in *Cxcr6*^−/−^ mice [123].

The literature on chemokines and their receptors in the field of human CVD is expanding. Analytical tools, such as gene expression microarray and RNA sequencing, allow for the detection of alterations in gene expression in different stages of the atherogenic process. Publicly available atherosclerosis related datasets are being benefited from and knowledge has been gained on the changes in the gene expression of chemokines and chemokine receptors [124]. Accordingly, CCL5, CCL19 and CXCR4 were identified as being more highly expressed in human plaque specimens as compared to control tissue [124]. Furthermore, the expression of CCL2 and CCR2 was observed to be increased in atheroma plaques as compared to distant macroscopically intact tissue from the same patient (GSE43292, [125]) while expression levels of CCL2 or CCR2 did not vary between an atherosclerotic and nonatherosclerotic arterial wall (as distal internal mammary artery) (GSE40231 [126]). The latter might be attributed to alterations in expression of some chemokines during particular stages of atherogenesis. To illustrate, multiple chemokines (CCL3, CCL5, CCL19, CCL21, CXCL12, CXCL16) and chemokine receptors (CCR1, CCR2, CCR5, CCR7, CXCR4, CX3CR1) were enriched in advanced atherosclerotic lesions as opposed to plaques of an early atherosclerotic stage (GSE28829, [127]). A recent study identified genes involved in atherosclerotic plaque progression, which could serve as novel biomarkers and eventually utilized as targets to get a grasp on the plaque development. Six core genes were identified (*FPR3, MS4A4A, C1QB, CCL18, CXCR4,* and *CXCL2*) in the protein−protein interaction (PPI) network, three of which (*CCL18, CXCR4*, and *CXCL2*) were markedly enriched in the chemokine signaling pathway and in particular the *CXCR4* and *CXCL2* genes could be linked to atherosclerotic plaque progression (GSE28829 and GSE41571, [128]). Furthermore, the chemokines discussed above, CCL5, CCL19 and CXCR4, were also more present in advanced plaques versus early-stage atherosclerotic plaque [124]. Of interest, chemokine or chemokine receptor alterations during atherogenesis seem more relevant in the plaque, since no differences were observed in peripheral blood mononuclear cell (PBMC) and peripheral blood samples (GSE20129, [129]). Altogether, these gene expression analyses reveal significant and interesting changes of chemokine and chemokine receptor expression in plaque as well as plaque progression. However, associations do not inevitably mirror causality, hence mechanistic studies are necessary to linking cause and consequence.

To summarize, chemokines and chemokine receptors are implicated in the orchestration of the various responses that lead to the progression of atherosclerotic lesions. These chemokine-receptor interactions have, as discussed, various roles in cell survival, regeneration, proliferation, and even the type of inflammatory response that ensues. Through these interactions, a pleiotropy of chemokine-receptor interactions are able to regulate the progression of atherosclerosis.

### 2.3. Regression in Atherosclerosis

Chemokines can also play a part in the amelioration of atherosclerosis. In this section, we will focus on chemokines and chemokine receptors which are involved in the regression of atherosclerosis.

One of the main chemokine receptors that has been implicated in atherosclerosis regression is CCR7. For example, when compared to healthy aortic tissue, an inverse correlation with atherogenesis was observed upon a downregulation of CCR7 expression in human carotid atheroma [130]. This observation was further supported by a mouse model by Wan et al., in which significantly increased lesion sizes were found in *Apoe*^−/−^*Ccr7*^−/−^ mice fed with a high-fat Western diet as opposed to control mice [131]. In addition, irradiated *Apoe*^−/−^*Ccr7*^+/+^ mice that received bone marrow from *Apoe*^−/−^*Ccr7*^−/−^ mice developed significantly larger atherosclerotic lesions in the aorta as compared to mice which received *Apoe*^−/−^*Ccr7*^+/+^ bone marrow [131]. Other studies also point to CCR7 as a key receptor in the emigration of macrophages from plaque, thus leading to a regression in plaque size [12]. Furthermore, in an environment of atherosclerotic regression, an increase in CCR7 mRNA and protein expression by foam cells was observed, while regression was hindered by the inhibition of CCR7 [132]. Therefore, these studies demonstrate that myeloid CCR7 not only has an atheroprotective role but is also involved in the reduction of established plaques through the egress of macrophages (Figure 2 Pathway 5). The use of the Reversa mouse model, which is a model that is widely used to study atherosclerosis regression, also pointed to the crucial role of CXCR2 and CX_3_CR1 in lesion regression. CXCR2 is of great importance in the homing of endothelial progenitor cells towards the injured vasculature. Meanwhile CX_3_CR1 can partly control the regenerative capacity of endothelial progenitor cells [133]. Additionally, endothelial progenitors have a role in the stabilization of plaques, thereby implicating these receptors in plaque stabilization [133].

The data and general consensus are quite clear that chemokines are heavily implicated in the initiation, progression, and potentially the regression of atherosclerosis. Utilizing the knowledge of these signaling pathways, their targets, and their consequences allows for various potential therapies in the treatment of atherosclerosis.

## 3. Targeting Chemokines and Chemokine Receptors: Therapeutic Potential in Atherosclerosis

Chemokines and their receptors are known to play a significant role in the development and progression of atherosclerosis and CVDs. In this section, we will address some current approaches that benefit from the role of chemokines and their receptors to intervene in the initiation and progression of atherosclerosis.

### 3.1. Preclinical Studies: Discovering the Therapeutic Potential of Chemokines and Chemokine Receptors

Various preclinical studies have exploited the knowledge of chemokine and chemokine receptor interactions in order to halt the progression or even reverse the effects of atherosclerosis. In this section, we will highlight some studies that target these interactions in the treatment of atherosclerosis and CV related complications.

One such strategy was evaluated by Winter et al. Their study pin-pointed the rhythmic nature of CCL2 release, allowing them to design a time-based pharmacological treatment which targets the chemokine receptor at the peak time of CCL2 mediated leukocyte recruitment [63]. Using a small-molecule CCR2 antagonist, RS102895, they showed that CCR2 neutralization, within the identified time-window, results in reduced monocyte recruitment without influencing the counts of circulating myeloid cells. Consequently, the use of this antagonist seems to have a positive effect and hampers the formation of atherosclerotic lesions. Although the success of this strategy shows a lot of potential for translation into humans, the circadian nature of CCL2 in humans would first have to be investigated in greater detail [63]. In contrast, ACKR3 was targeted with an agonist by Li et al. [117]. They evaluated the use of the synthetic ligand CCX771, which prevents the binding of CXCL12 to ACKR3 and can activate the cholesterol-lowering effects regulated by ACKR3 [117]. CCX771 injections in atherosclerotic *Apoe*^−/−^ mice resulted in decreased hyperlipidemia, monocytosis, and the reduced formation of atherosclerotic lesions in aortic roots as compared to mice treated with vehicle [117]. The studies above showed promising results; however, the usage of competitive ligands or blocking of these receptors must be carefully scrutinized due to the complexity of their roles and their physiological function in important immune responses.

Another strategy to prevent the progression of atherosclerosis could be the inhibition of the proatherogenic chemokine receptor CCR5. The CCR5 antagonist retroviral drug, Maraviroc, was used in two murine models, which represented two different stages of atherosclerotic progression [134]. The first model evaluated the effect of Maraviroc in ritonavir-induced atherosclerosis in *Apoe*^−/−^ mice, depicting an early phase of atherosclerosis. Maraviroc reduced atherosclerotic plaque size as well as the lesional macrophage content. The second model, representative of a more advanced stage of atherosclerosis, also examined the effects of Maraviroc treatment in *Apoe*^−/−^ mice. Here, atherosclerotic progression was inhibited by a reduction of macrophage infiltration [134]. These results suggest that CCR5 is a promising therapeutic target at varying stages of atherosclerotic disease progression.

Many strategies relied on receptor antagonism to counteract the role of chemokines and chemokine receptors in atherogenesis; however, over the last decade, the role of microRNA (miRNA) in regulating various key pathophysiological processes of atherosclerosis has come to light [135]. MiRNAs were linked to regulating lipid homeostasis, cellular adhesion, and even the generation of key inflammatory mediators such as chemokines [135]. This knowledge led to studies focusing on different miRNAs in an attempt to prevent and regress atherosclerosis [135,136]. A prime example is the study by Ma et al., where chemokines like CCL2, CCL5, CCL8, and CXCL9 were suppressed in endothelial cells by delivering miRNA-146a/-181b to the cells using E-selectin-targeting multistage vector microparticles (ESTA-MSV) [137]. Additionally, the aortic areas of *Apoe*^−/−^ mice fed with a Western diet showed reduced plaque sizes, as well as a reduction of lesional macrophages and chemokine expression while demonstrating more stable plaques when treated with the miRNAs as opposed to controls [137].

Furthermore, literature searches revealed various other studies that focused on the amelioration of atherosclerosis and related disorders by disrupting chemokine signaling, which focused for example on CX_3_CR1, MIF, CXCR3, and the CCL5-CXCL4 heterodimer (Table 1).

### 3.2. Clinical Trials: Targeting Chemokines and Their Receptors as Therapeutic Tools

Treatments targeting chemokine receptors, such as CCR5 and CXCR4, are already approved for clinical use to treat, for example, human immunodeficiency virus (HIV) and some cancers. Interestingly, these receptors are also heavily implicated in atherogenesis [138,139,140]. Therefore, the same strategies can be utilized in the context of atherosclerosis. Nowadays, chemokines and their receptors are evaluated in a myriad of studies and reported in the ClinicalTrials.gov (accessed on 22 July 2021) and the European Union Clinical Trials Register databases. Some are potential therapeutic targets for various disorders that, similar to atherosclerosis/CVDs, have an underlying inflammatory component [141,142,143]. The trials specifically focusing on CVDs and CVD-related complications are summarized in Table 2.

Following the success of the CCR5 antagonism in preclinical studies, Maraviroc was evaluated for the treatment of atherosclerosis in a clinical trial of HIV patients with increased CV risk [162]. The drug successfully lowered various markers of CV risk, yet it did not influence plasma cholesterol levels, systemic inflammation, and immune activation, which could otherwise result in an increased risk of severe infections. These very promising results could translate to the clinical use of Maraviroc as an anti-atherosclerotic drug, although the small sample size of 21 patients calls for further investigation [162]. Another study directly evaluated the effects of a CCR2 antagonist, MLN1202, on atherosclerotic risk [161]. This phase 2 trial observed a decrease in CRP levels in serum upon MLN1202 treatment [161], pointing towards a reduced risk in the development and progression of atherosclerosis. Additionally, several trials investigated the effect of the direct inhibition of chemokines. For example, an inhibitor of MCPs, such as CCL2, CCL7, CCL8, called Bindarit, was tested in the context of coronary restenosis following a percutaneous coronary intervention. Although the primary endpoint of in-segment late loss (of lumen size) was not met, a reduction in in-stent late loss was observed, demonstrating its potential for clinical application [163]. Although these trials yielded promising results, further validation is still needed to reach the next step to become an approved therapy targeting chemokines or chemokine receptors.

A different approach utilized a nonviral DNA plasmid of CXCL12 called JVS-100 [164]. This strategy focused on repairing damaged tissue, rather than attempting to decrease atherosclerotic burden or CV risk. Interestingly, CXCL12 has been shown to be an important promoter of tissue repair. Overexpression of CXCL12 induced homing of cardiac stem cells to the sites of ischemic tissue damage and inhibited apoptosis of cardiac myocytes, therefore being an appropriate strategy for dealing with heart failure and various ischemic injuries [164]. One trial involved endomyocardial injections of JVS-100 and observed an improvement of composite scores following JVS-100 treatment. Unfortunately, the primary endpoint of this study could not be met [164], and further investigation is required to define the exact effect of this treatment on CV risk.

Most of these trials revealed that targeting chemokines and their receptors is a promising, though still challenging, therapeutic approach. Many more compounds intervening with chemokines and chemokine receptors were tested in preclinical settings in the context of CVDs, indicating a potential for the increase of clinical trials evaluating chemokine and chemokine receptor interventions for the treatment of various CV complications and diseases in the near future [4].

### 3.3. Imaging as a Diagnostic Tool in Atherosclerosis

As described above, chemokines and chemokine receptors represent an attractive target for therapy. Additionally, several studies also demonstrate the possibility of exploiting these molecular targets as a diagnostic tool. Imaging modalities are great as diagnostic apparatus as they allow for the evaluation of plaque size and location. However, traditional imaging modalities have failed to elicit more information about plaque composition. Nevertheless, to visualize molecular activity in vivo, imaging modalities such as positron emission tomography (PET) [167], single-photon emission computed tomography (SPECT) [168] and the PET combined modality PET/magnetic resonance imaging (MRI) can be used [169]. These imaging techniques provide additional information on plaque localization and composition, which can result in the development of more target-specific treatments [167]. For example, Li et al. used PET/MRI imaging to evaluate the presence of CXCR4, which colocalized with the macrophages found in atheromas. In atherosclerotic lesions, an increase of a radiolabeled chemokine receptor ligand, Gallium-68 (^68^Ga) Pentixafor, which has a high affinity for CXCR4, was detected. In addition, in patients with an increased CV risk, a significantly higher uptake of the radiolabeled chemokine was observed in atherosclerotic lesions [169,170]. Another tracer, developed for PET imaging, is based on viral macrophage inflammatory protein 2 (vMIP-II), a ligand which is able to bind to multiple chemokine receptors, including CCR2, CCR3, CCR5, ACKR3, CX_3_CR1, and CXCR4 [171]. This vMIP-II-based PET nanoprobe, known as ^64^Cu-DOTA-vMIP-II, was revealed to be targeting endothelial CXCR4 in *Apoe*^−/−^ mice. In addition, studies with ^64^Cu-DOTA-vMIP-II demonstrated an increased CXCR4 expression on plaque margins experiencing a high degree of endothelial dysfunction, thus exposing this tracer as a diagnostic tool for injuries to the endothelium [171]. Other chemokine receptors exploited for imaging of atherosclerotic lesions include CCR5 and CCR2 [172]. Wei et al. established the CCR5-specific SPECT tracer ^111^In-DOTA-DAPTA [168]. In vitro results showed a higher uptake of the tracer in cells expressing high CCR5 levels, which was corroborated ex vivo as ^111^In-DOTA-DAPTA lesion uptake in *Apoe*^−/−^ mice was significantly higher than in C57BL/6 mice. In addition, preinjections with DAPTA, a CCR5 binding peptide, decreased the uptake of ^111^In-DOTA-DAPTA, thereby showing that ^111^In-DOTA-DAPTA specifically targets CCR5 [168]. In the case of CCR2, a clinical trial recently began recruiting participants and aims to evaluate arterial atherosclerosis using the imaging agent ^64^Cu-DOTA-ECL1i (ClinicalTrials.gov (accessed on 22 July 2021) Identifier: NCT04537403).

Safe usage of these promising modalities has to be considered. These diagnostic tools should not cause side effects, and controlled biodegradability and rapid elimination from the body have to be ensured. Before new modalities can be safely translated into the clinic, fundamental steps concerning the pharmacokinetics are required. Specificity is a main criteria in this new area of diagnosis [173] and interference of the modalities with the general inflammatory process in the body should be avoided. A limitation of nuclear imaging is its resolution of only a few millimeters. Therefore, PET or SPECT, are often performed simultaneously with a CT-scan to localize the accumulation of the radionuclides in the body in two or three dimensions. This results in the exposure of patients to supplementary ionizing radiations [173].

When safety can be assured, information on plaque localization and molecular activity will facilitate the development of more target-specific treatments [167]. These imaging modalities can act not only as tools for diagnostic and for the delivery of targeted treatments, but can also improve our understanding of the role chemokine receptors play throughout the development of atherosclerosis, which would inevitably lead to improved therapies.

## 4. Concluding Remarks

Although chemokines and their receptors are considered to play a significant role in the development and progression of atherosclerosis and thus CVDs, the appropriate clinical treatments have yet to be approved [4]. The complexity of the interactions between these molecules and receptors makes it an incredible challenge and, despite the attractiveness of chemokines and chemokine receptors as therapeutic targets, potential treatments are still under investigation and are not being clinically applied at this time. The general lack of accepted therapies is caused by a variety of reasons, including problems like the intrinsic redundancy of chemokines and their receptors, insufficient knowledge of the mechanistic roles of receptors and chemokines in disease, the potential to block important modulators of protective and necessary immune responses [174], the inadequate selection of dosage and targets, and the circadian-dependent effect of interventions [175]. Consequently, antagonism and deletion of genes for certain chemokines or their receptors has quite a complex effect. Furthermore, contradicting results suggest that certain chemokines may play a role at different stages of the development of atherosclerosis. All of these factors present extraordinary hurdles that must be cleared in the race to develop therapies targeting chemokines/chemokine receptors. However, these challenges have not deterred scientists from attempting to unlock the key to targeting chemokines. Increasing interest continues to result in even more knowledge and promising approaches. In particular, the increase of successful results from preclinical trials suggests future treatments targeting these molecular signaling pathways are on our horizon. Additionally, the plethora of clinical studies being done on chemokines in the settings of other inflammatory diseases could indicate a potential translation into the field of atherosclerosis and CVDs [4]. Once safety concerns and patient tolerance are evaluated, the compounds tested in different disease settings could be considered and their effect tested in atherosclerosis. Clearly, these emerging therapies using miRNAs, rhythmicity in chemokine release, peptides inhibiting heterodimerization, and various other tools to modulate chemokine receptor activation via small-molecule agonists and antagonists, point to a bright future in the search to deliver a treatment with the potential to ameliorate or stop atherosclerotic disease progression.

## Figures and Tables

**Figure 1 jcm-10-03825-f001:**
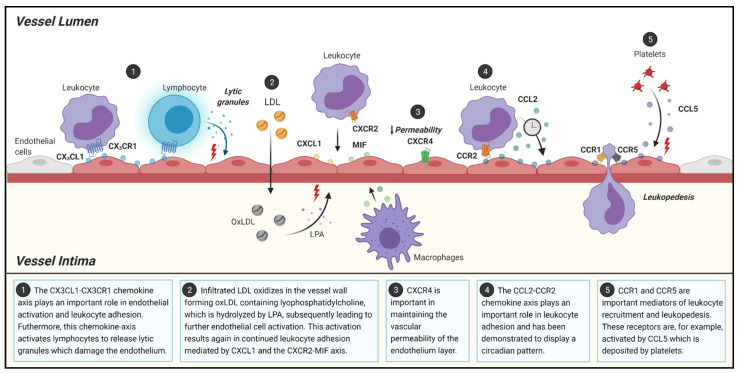
Schematic overview of the role of chemokines and chemokine-receptors in different processes during atherosclerosis initiation. LDL: low density lipoprotein; LPA: lysophosphatidic acid. Figure was created using BioRender.com.

**Figure 2 jcm-10-03825-f002:**
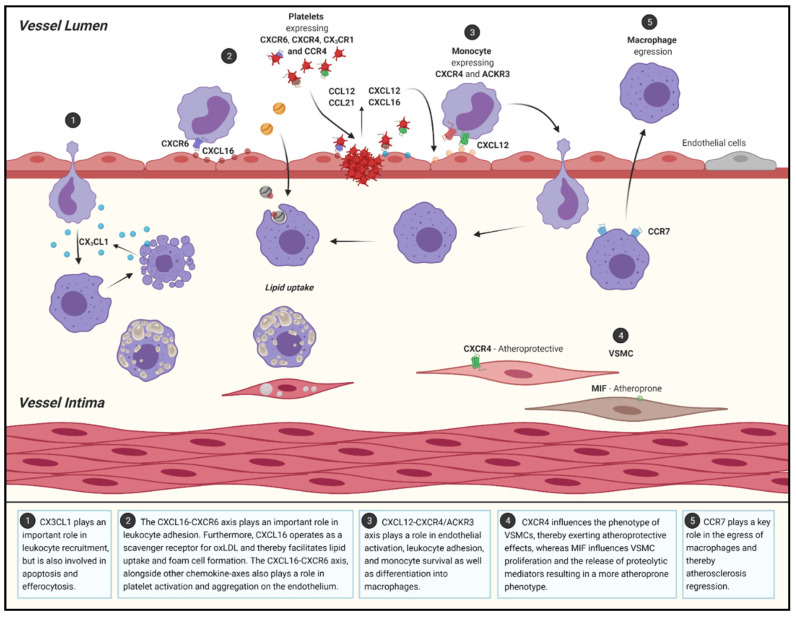
Schematic overview of the role of chemokines and chemokine-receptors in different processes during atherosclerosis progression and regression. VSMC: vascular smooth muscle cells. Figure was created using BioRender.com.

**Table 1 jcm-10-03825-t001:** Preclinical studies targeting chemokines and chemokines receptors in mice.

Target	Treatment	Type of Treatment	Condition	Outcome	Reference
ACKR3	CCX771	Small-molecule receptor agonist	Atherosclerosis	↓Lesion formation ↓Blood cholesterol	[117]
	TC14012	Small-molecule receptor agonist	Acute myocardial infarction	↓Infarct size Only [144]: ↑Left ventricular internal diameter ↑Left ventricular volume ↑Vascular density	[144,145]
CCR2	RS102982	Small-molecule receptor antagonist	Atherosclerosis	↓Lesion size ↓Macrophage accumulation	[63]
	Nanoparticle encased siRNA	Small silencing RNA	Myocardial infarction	↓Ly6C^hi^ monocytes recruitment ↓Post-MI heart failure ↓Left ventricular dilation	[146]
	Nanoparticle encased siRNA	Small silencing RNA	Myocarditis	↓Ly6C^hi^ monocytes in heart ↓Myeloid progenitor trafficking Improved ejection fraction	[147]
CCL2, CCL5, CCL8, CXCL9	miR-146a/-181b packaged in ESTA-MSV microparticles	microRNA delivery	Atherosclerosis	↓Plaque size ↓Macrophage accumulation ↑VSMCs in plaque ↑Collagen in plaque	[137]
CCR5	Maraviroc	Small–molecule receptor antagonist	Atherosclerosis	↓Plaque size ↓Macrophage infiltration	[134]
	[(44)AANA(47)]-RANTES	Chemokine receptor antagonist (modified chemokine)	Myocardial ischemia and reperfusion	During early myocardial reperfusion: ↓Infarct size ↓Myocardial leukocyte infiltration ↓Oxidative stress ↓Apoptosis	[148]
	Met-RANTES	Chemokine receptor antagonist (modified chemokine)	Atherosclerosis	↓Lesion size ↓Leukocyte infiltration ↑Collagen content in atheroma	[54]
CCR5, CXCR3	TAK-779	Small–molecule receptor antagonist	Atherosclerosis	↓Lesion formation ↓T-helper 1 cell plaque infiltration	[149]
CCL5/CCL17	CAN peptide	Inhibition of heteronomer formation	Atherosclerosis	↓Lesion size in aortic root	[150]
CCL5/CXCL4	MKEY peptide	Inhibition of heteronomer formation	Atherosclerosis	↓Lesion formation ↓Macrophage accumulation	[151]
CCL5/HNP1	SKY peptide	Inhibition of heteronomer formation	Myocardialinfarction	↓Ly6C^hi^ monocyte adhesion/recruitment ↓Macrophages ↓Inflammation	[152]
CCL5,CCL11 CXCL1	Evasin-4 (CC-) Evasin-3 (CXC-)	Chemokine- binding protein (inhibits chemokin binding)	Myocardial infarction	↑Improved survival (Evasin-4) Post infarction: ↓Leukocyte infiltration ↓ROS production ↓Neutrophil chemoattractants ↓Infarct size	[153]
CXCL1, CXCL2	Evasin-3	Chemokine-binding protein (inhibits chemokin binding)	Myocardial ischemia	↓Infarct size In infarcted myocardium: ↓Neutrophil infiltration ↓ROS production	[154]
CCL2, CCL5, CX_3_CL1	M3	Chemokine-binding protein (inhibits chemokine binding)	Atherosclerosis	12-week model: ↓Lesion area ↑Aortic smooth muscle α-actin expression 6-week model: ↓Macrophage content in plaques ↓Lipid deposition in thoracic aorta	[155]
CXCR3	NBI-74330	Small–molecule receptor antagonist	Atherosclerosis	↓Lesion formation ↓Leukocyte migration Improved regulatory/effector T cell balance	[156]
	AMG487	Small–molecule receptor antagonist	Cardiac remodeling	Abrogation of ↑ in ventricle weight to body weight ratio ↓Macrophage recruitment ↓Cardiac remodeling	[157]
CX_3_CR1	F1	Chemokine receptor antagonist (modified chemokine)	Atherosclerosis	↓Monocyte adhesion ↓Macrophage accumulation in aortic sinus ↓Monocyte survival ↓Lesion size in advanced atherosclerosis	[158]
	BI 655088	Variable domains of camelid heavy chain-only antibody (antagonist)	Atherosclerosis	↓Lesion formation	[159]
MIF	COR100140, anti-MIF monoclonal antibody	Small–molecule receptor antagonist	Myocardial infarction	COR100140: ↓Incidence of cardiac rupture Antibody: ↓CCL2 expression ↓Leukocytes at infarct region at 24 h	[160]

↓: Indicates decrease ↑: Indicates increase.

**Table 2 jcm-10-03825-t002:** Clinical trials registered on clinicaltrials.gov (accessed on 22 July 2021) and clinicaltrialsregister.eu focusing on treating CVDs by targeting chemokines and chemokine receptors.

Target	Intervention	Aim	Condition	Phase	Status and Results	Trial Identifier
CCR2	**MLN1202** humanized monoclonal antibody	Measuring the effects of MLN1202 on C-reactive protein levels in patients with risk factors for CV disease	Atherosclerosis	II	Completed; well tolerated in patient population and significant reduction in high-sensitivity C-reactive protein levels [161]	NCT00715169
CCR5	**Maraviroc** Small-molecule receptor antagonist	Augmenting rehabilitation outcomes after stroke	Stroke	II	Not yet recruiting	NCT04789616
	**Maraviroc** Small-molecule receptor antagonist	Efficacy of Maraviroc in modulating atherosclerosis in HIV patients	Atherosclerosis	IV	Significant improvement in various markers of CV risk including carotid atherosclerosis, endothelial dysfunction, and arterial stiffness. No effect on systemic inflammation apparent [162]	NCT03402815
CXCR2	**AZD5069** Small-molecule receptor antagonist	Evaluate inhibition of CXCR2 as a treatment of coronary heart disease	Coronary heart disease	II	Ongoing	EudraCT 2016-000775-24
CXCR4	**POL6326** Peptidic receptor antagonist	Evaluate effects of CXCR4 inhibition in patients with large reperfused ST elevation myocardial infarction	Large reperfused ST-elevation myocardial infarction	II	Completed; results not found	NCT01905475
CCL2	**Bindarit** Selective inhibitor	Evaluating the efficacy and safety of different Bindarit dosages in preventing stent restenosis	Coronary restenosis	II	Primary endpoint not met. Reduction in the in-stent late loss observed. Bindarit was well tolerated [163]	NCT01269242
CXCL12 (SDF-1)	**JVS-100** nonviral DNA plasmid (transient CXCL12 expression)	Evaluate the safety and efficacy of a single dose of JVS-100 administered by endomyocardial injection to cohorts of adults with ischemic heart failure	Ischemic heart failure	II	Primary endpoint was not met. Safety profile supports repeat dosing with plasmid SDF-1. Potential for attenuation of left ventricular remodeling and improvement of ejection fraction [164]	NCT01643590
	**JVS-100** nonviral DNA plasmid (transient CXCL12 expression)	Evaluate the safety and efficacy of JVS-100 administered by retrograde delivery to cohorts of adults with ischemic heart failure	Ischemic heart failure	I/II	Unknown	NCT01961726
	**JVS-100** nonviral DNA plasmid (transient CXCL12 expression)	Evaluate the safety and efficacy of JVS-100 administered by direct intramuscular injection to cohorts of adults with critical limb ischemia	Critical limb ischemia	II	Completed; results not found	NCT01410331
	**JVS-100** nonviral DNA plasmid (transient CXCL12 expression)	Evaluate the safety and efficacy of JVS-100 administered by direct intramuscular injection as adjunct to revascularization of infrapopliteal lesions in patients with advanced peripheral artery disease and tissue loss	Peripheral arterial disease	II	Primary efficacy endpoint was not met at either 3- or 6-month follow-up, intervention failed to improve wound healing [165,166]	NCT02544204
	**ACRX-100** nonviral DNA plasmid (transient CXCL12 expression)	Evaluate the safety of a single escalating dose of ACRX-100 administered by endomyocardial injection to cohorts of adults with ischemic heart failure	Heart failure	I	Completed; results not found	NCT01082094

## Data Availability

Not applicable.

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
