# Peer review of "Key Chemokine Pathways in Atherosclerosis and Their Therapeutic Potential"

_jcm, 2021, doi:10.3390/jcm10173825_

Round 1

Reviewer 1 Report

The current review article "Therapeutic potential of chemokines in atherosclerosis" is mainly discussed the importance and the role of chemokine and chemokine receptor axis involvement during initial and during atherosclerosis development. the authors have well written the manuscript.

Though the authors have discussed all the major literature in this field, the reviewer has few concerns.

  1. Though, the authors mainly focused on chemokine and chemokine receptor's role in atherosclerosis plaque region only and initial stages of the disease progression. The title of the article is very general and not very specific. The reviewer thinks, that It would be important to discuss the role of chemokines in the advanced stages too. Since chemokines are involved in immune cell recruitment, the authors need to discuss the chemokine's role in immune cell infiltration into the adventitia during early and advanced atherosclerosis (TLO formation). TLOs were observed in both humans and experimental mice.
  2. Since B cells play a very important role in disease modulation, what are the chemokines/receptors involved in B cell recruitment to diseased aorta?
  3. Since perivascular fat directly connects with aorta adventitia, secrete chemokines, harbors high numbers of immune cells (form FALCs), and PVAT directly regulates atherosclerosis development, it would be more informative to discuss the role of chemokines in perivascular fat and its regulation on atherosclerosis progression. 
  4. the authors need to discuss the literature on chemokines and their receptors data in the field of human CVD.

Reviewer 2 Report

The paper entitled “ Therapeutic Potential of Chemokines in Atherosclerosis” reviews current knowledge about the role of chemokines in atherosclerotic plaque progression on different stages with additional report about potential therapeutic agents interfering with chemokine networks in both pre-clinical and clinical studies. In general, it’s interesting comprehensive review which describes the most important aspects of the topic. Hovewer, there are some shortcomings which should be corrected before the acceptance for the publication in JCM.

Major faults:

  1. In the “Introduction” section (Atherosclerosis - 1.1.) the Authors concentrate upon lipid interplay in atherosclerotic plaque formation, with only general information about the role of immune system in the process of atherogenesis. There is a lack of information about the most important components of the immune system involved in this process (in particular – there is no information about nuclear factors like NF-κB, which are transcription factors for many chemokines described in the manuscript).
  2. Although chemokines, their receptors, interactions and involvement in the process of atherogenesis have been mentioned, enumerated and described, the crucial pathways have not been highlighted. I suggest either present the most important chemokine-receptor interactions on Figures (after rearrangement of Figures – according to the point 3) or on additional Tables.
  3. Figures don’t correspond with the text of the manuscript. They should be put after the entire sections (e.g. Figure 1 after the Section 2.1) as a summary. Moreover, the processes marked on Figures as numbers in black circles should not be described in the text below the Figure legend. The description of processes should be rewritten in a more compact way and put on Figures (even as a frame below the drawing, but as an integrated element of the Figure).

Minor faults:

  1. The information highlighting the role of chemokines as crucial factors in the atherogenesis should be introduced (e.g. like in article - Yang R, Yao L, Du C, Wu Y. Identification of key pathways and core genes involved in atherosclerotic plaque progression. Ann Transl Med. 2021 Feb;9(3):267).
  2. In the section 3.3 “Imaging as a diagnostic tool in atherosclerosis” a piece of information about safety concerns in molecular imaging should be included (as it was pointed out for example in in: Juenet M, Varna M, Aid-Launais R, Chauvierre C, Letourneur D. Nanomedicine for the molecular diagnosis of cardiovascular pathologies. Biochem Biophys Res Commun. 2015 Dec 18;468(3):476-84).
  3. Short information about the directions of further research in the field chemokines in atherogenesis should be introduced .
  4. In the Table 1 and Table 2 the direct information whether a certain chemokine is up- or down-regulated should be introduced.
  5. I suggest that the entire text should be checked by English-language native speaker or aglicist. There are linguistic faults or imprecise expressions – as in the examples below:
    1. Line 37 – “The leading cause of CVDs has been recognized to be atherosclerosis”- improper style
    2. Line 69 – “(… )and accumulation of lipids, such as low-density lipoprotein (LDL).” – imprecise information, lipoproteins are not lipids.
    3. Line 91 – “The accumulation of these lipid-rich cells in the intima results in a so called fatty streak, which following further cell recruitment develops into an atheromatous plaque [26].” – “results in generation of fatty streak”, moreover the syntax of this sentence is wrong.
    4. Line 123 – “The ligands and their receptors are known to be highly promiscuous regarding selectivity” – this sentence should be rewritten, the word “promiscuous” is unsuitable in this context.
    5. Line 136 – “As described above, atherosclerosis begins following dysfunction of the arterial endothelium” – improper syntax of the sentence
    6. Line 139 – “The notable chemokines and chemokine receptors involved in the initiation of atherosclerosis will be highlighted in this section” – Present Simple tense, not Future Simple is appropriate.
    7. Line 318 – “However, certain VSMCs phenotypes are associated with atherogenesis. VSMCs deficient in CXCR4 exhibit a contractile phenotype which is connected to the development of atherosclerosis [44], demonstrating a potential atheroprotective effect of SMC-expressed CXCR4”. - even though this sentence provides correct information, it’s misleading. First part of the sentence “(…) CXCR4 exhibit a contractile phenotype which is connected to the development of atherosclerosis” wrongly suggests, that contractile phenotype is pro-atherogenic.

Round 2

Reviewer 1 Report

The reviewer did not have any further concerns.

Author Response

We would like to thank the reviewer for his/her positive feedback on our revised manuscript. 

Reviewer 2 Report

The Authors have modified the manuscript according to my suggestions. It’s now improved and refined, therefore in my opinion it is now worth of publication.

Currently, there are only some minor faults in the entire manuscript.

Minor linguistic mistakes

  • Line 42: “incidences” – it should be “incidence”
  • Line 84: between “atherosclerosis” and “thereby” a comma is needed.
  • Line 119-123: “In a pro-inflammatory environment, such as in atherosclerotic lesions, this process is disturbed due to the expression of cholesterol transporters and NCEH being downregulated and the amount of oxLDL as well as the expression of ACAT1 is  upregulated, leading to the excessive accumulation of lipid droplets in the endoplasmic reticulum, which in turn results in the formation of foam cells” – this sentence is too complex, it should be rewitten (from grammatical point of view).
  • Line 173: „(…) exhibit a more pro-inflammatory phenotype” – it should be written without „a”
  • line 338: „stromal cell derived factor 1” – it should be „stromal cell-derived factor 1”
  • Line 398: a comma after „(…) in CXCR6,” is unnecessary.

Author Response

We would like to thank the reviewer for his/her positive feedback on our revised manuscript.

We have adjusted the remaining minor linguistic issues as suggested.